# Meta-analytical approaches to ordinal outcome data in clinical interventional studies: A scoping review with reproducible research

Ali Mulhem[1,2]*

**1** Westfälische Wilhelms-Universität Münster, Münster, Germany, **2** Department of Continuing Education, DPhil Program in Evidence-Based Health Care, & Kellogg College, University of Oxford, Oxford, United Kingdom

* ali.mulhem@kellogg.ox.ac.uk

## Abstract

### Background

Conducting meta-analyses on ordinal outcome data is more complex than on binary or continuous data. This study aims to summarise the current biomedical literature on meta-analytical methods for ordinal outcomes and attempts to reproduce the results of previous studies.

### Methods

A systematic search was conducted in three databases, MEDLINE, EMBASE, and PsycINFO, from inception to 05/05/2024. Forward and backward citation searches were also performed. The screening was conducted in two phases using Covidence software. Studies were included if they reported or compared methods for meta-analysis of ordinal outcomes in clinical interventional studies. Relevant studies were summarised and discussed. If sufficient data for methods comparison were available, either in the retrieved reports or after contacting the authors, an attempt at reproducible research was made.

### Results

333 records were screened, yielding four methodological studies that met the inclusion criteria. These studies addressed meta-analytical methods for ordinal scales ranging from 5 to 20 ordered categories. The three primary approaches identified were (1) ordinal models (proportional odds and generalised odds), (2) binary models (dichotomisation of ordinal scales), and (3) continuous models (treating ordinal scales as continuous variables). None of the included studies provided a comprehensive comparison of all three approaches. Two studies compared different proportional odds models; one compared binary with proportional odds, but the full results were not published, and one compared continuous and generalised odds models using simulated data with one scenario. The latter study allowed

**Data availability statement:** All relevant data are within the manuscript and its Supporting Information files.

**Funding:** The author(s) received no specific funding for this work.

**Competing interests:** The authors have declared that no competing interests exist.

for reproducibility, but our analysis produced different results, and attempts to clarify the discrepancy with the authors were unsuccessful.

## Conclusions

A significant knowledge gap exists regarding the optimal meta-analytical method for ordinal outcomes in clinical interventional studies. Further methodological research is required to establish a robust evidence base for choosing the most appropriate approach.

## Background

Meta-analysis is a vital research method used to estimate the effect sizes of clinical interventions [1,2]. The statistical methods employed in meta-analyses vary depending on the type of outcome data [3]. For binary and continuous data, robust and well-documented techniques are available [1,3]. However, in many biological and medical research fields, response variables are frequently measured as ordinal data [4], typically scales or scores that categorise variables into ordered groups [4]. Although ordinal scales are widely used in clinical research, meta-analytical methods for these data types are less straightforward than binary or continuous data [2].

Meta-analysing ordinal data is complex due to challenges in selecting appropriate summary statistics. Ordinal data straddle the line between qualitative and quantitative, lacking true metric properties despite numerical labels [4]. Unlike continuous or binary data, ordinal data require specific assumptions for meaningful summarisation [5]. Although recent studies have highlighted these challenges in the context of primary research [6,7], many randomised controlled trials (RCTs) continue to dichotomise ordinal outcomes or use suboptimal approaches without fully addressing their underlying assumptions. No comprehensive review have addressed this situation in the context of meta-analytic methodology for ordinal data in the biomedical literature, leaving a significant knowledge gap.

Given the widespread use of ordinal scales in clinical research—ranging from patient-reported outcomes to clinician-assessed measures—the lack of guidance on their meta-analysis poses challenges for synthesising evidence effectively [4,7,8]. For instance, dichotomising ordinal scales often results in a loss of information, reducing statistical power and potentially leading to biased estimates. Similarly, treating ordinal data as continuous may violate underlying assumptions and produce misleading conclusions. This creates an urgent need to evaluate and compare the methods available for meta-analysing ordinal data systematically.

This review summarises the current biomedical literature on meta-analytical methods for ordinal outcomes and explores the reproducibility of the reported results. By addressing this knowledge gap, the review aims to guide researchers in selecting appropriate statistical approaches, improving the accuracy and reliability of evidence synthesis in clinical research.

## Methods

### Scoping review

**Protocol and registration.** This scoping review was conducted as part of a DPhil thesis, registered with the Open Science Framework under https://doi.org/10.17605/OSF.IO/WJY5D. The scoping review is conducted according to the Preferred Reporting Items for Systematic reviews and Meta-Analyses (PRISMA) statement and its extension (PRISMA-ScR) [8,9].

**Eligibility criteria.** Eligible studies for this review compared or reported meta-analytical methods for ordinal outcome data in clinical interventional studies. We restricted the

inclusion to only methodological studies; we used the term methodological study to refer to any study that reports on the analysis of research-related reports (i.e., research-on-research) [10]. Thus clinical studies that used different meta-analytical methods to conduct the statistical analysis, not with the aim of methods comparison, were excluded. Studies focused on meta-analysis of non-interventional studies, such as diagnostic accuracy studies, were also excluded. There were no restrictions on publication language or date.

**Information sources.**  We restricted the search to three databases—MEDLINE, EMBASE, and PsycINFO—which were searched through the Ovid interface from inception to May 5, 2024. Additionally, forward and backward citation searches were performed. The search strategy comprised keywords and MeSH terms as follows: ((meta-analysis OR "exp Meta-Analysis") OR ("systematic review" OR "exp Systematic Review")) AND ("ordinal scale" OR "ordinal score" OR "ordinal data" OR "ordinal outcome").

**Selection of studies.**  After removing duplicates, one reviewer (AM) conducted a two-stage screening process through Covidence software [11], involving title/abstract screening followed by a full-text review.

**Data extraction and summary.**  The following data items were extracted by AM: study ID (first author and year of publication), sample size (number of studies included in the methodological analysis), the ordinal outcome scale, the type of data reported for the ordinal scale (raw data vs. summary statistics), methods of meta-analysis, metrics used to calculate the effect size, and, where available, the results of comparisons between different meta-analytical methods concerning the direction of the effect size.

## Reproducible research

Where sufficient data were available in the included studies or after contacting the authors, a reproducible analysis was conducted to replicate the original results. Statistical procedures were performed using RStudio (version 2022.12.0 + 353) [12] and STATA 18 BE Basic Edition [13]. I conducted reproducibility research as part of this review. The primary aim was to verify the findings of methodological studies identified during the review process. If these studies provided potential solutions to the challenges of meta-analysing ordinal data, confirming their results was essential to assess the reliability and applicability of their approaches. Reproducibility research not only enhances the credibility of the evidence but also identifies discrepancies and areas requiring further clarification, thereby advancing the methodological field.

## Results

### Scoping review

**Flow diagram.**  Three hundred thirty-three records were identified: 245 from the database search and 88 from citation searches (see Fig 1).

In the database search, after the removal of 90 duplicates, 155 records were screened by title and abstract. Of these, 134 records were deemed irrelevant and excluded. The remaining 21 records were assessed for eligibility, and 18 were subsequently excluded for the following reasons: studies focusing on statistical methods for ordinal data in primary research (i.e., not within a meta-analysis context; n = 14), studies reporting meta-analytical methods for ordinal data in diagnostic accuracy studies (i.e., not within an interventional context; n = 3), and studies focusing on statistical methods to compute correlations between variables (i.e., not effect size for interventions; n = 1).

In the citation search, 88 records were screened. One record could not be retrieved, and 86 records were deemed irrelevant and excluded. Lastly, one study was deemed eligible and included.

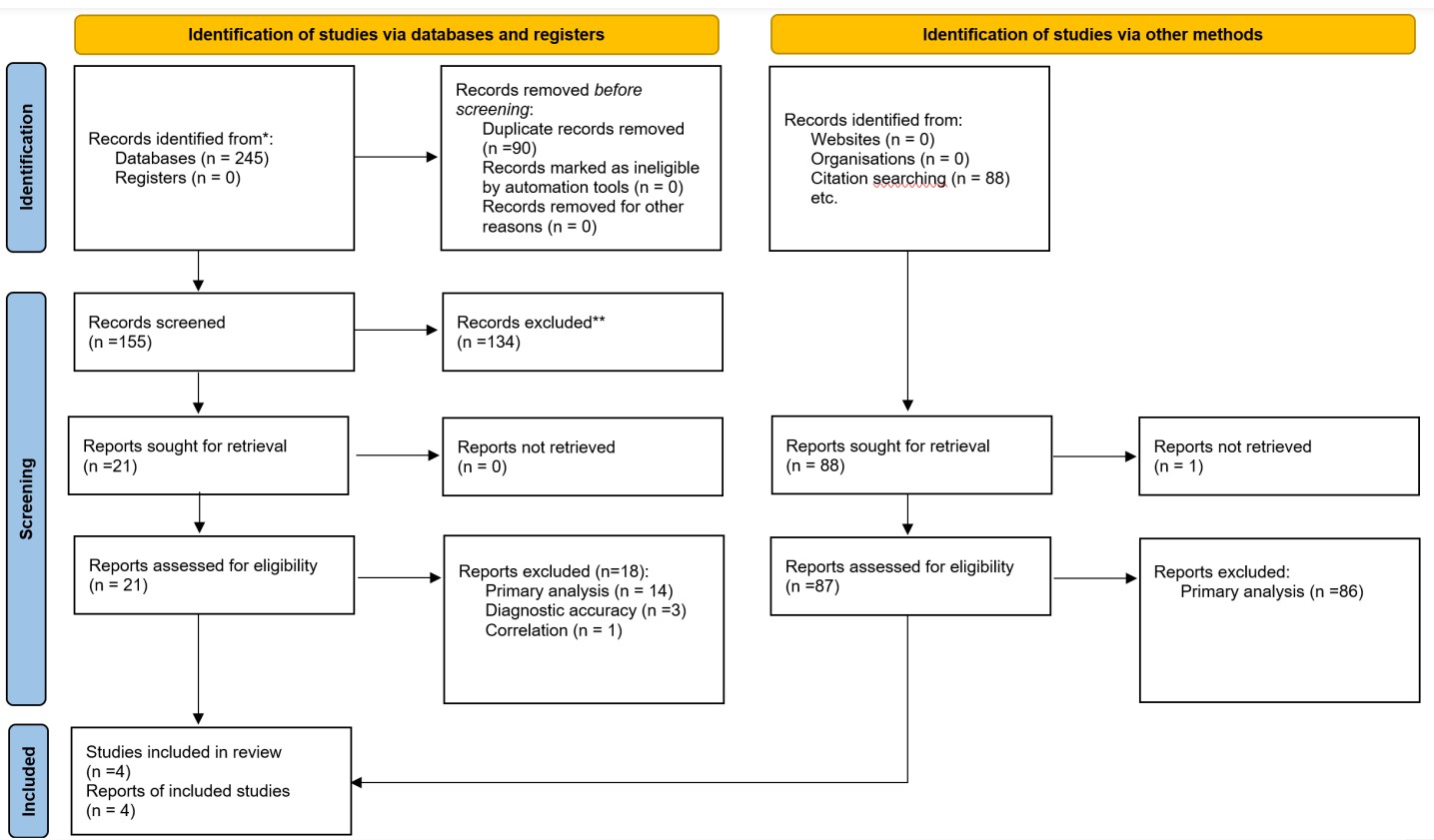

**Fig 1. PRISMA flow diagram of the scoping review.**

In total, we included four studies. Fig 1 illustrates this review's PRISMA 2020 flow diagram and the reasons for exclusion [8].

Among the four included studies, one was available only as a conference paper, obtained after contacting the first and last authors. Despite its limited information, it was included because it contained relevant data [14].

**Characteristics of included methodological studies.** All ordinal scales meta-analysed in the included studies had between 5 and 10 categories, with increments of either one or half points. So, the scales used in these studies had categories of 5 and 20. The number of meta-analysed studies (as samples) within each methodological study was relatively small, with counts of 5, 8, or 13. The exception was the conference paper, which analysed 216 studies across 24 meta-analyses but did not specify the number of studies in each analysis [14].

Table 1 provides a summary of the characteristics of the included studies.

**Methods to meta-analyse ordinal data.** Three primary approaches for meta-analysis of ordinal data were identified. The first is the **continuous approach**, where the ordinal scale is treated as a continuous variable to calculate the effect size, using the standardised mean difference (SMD) as a metric. The second is the **binary approach**, where the ordinal scale is dichotomised at a specified cut-point to generate a binary variable, with the effect size calculated using the logarithmic odds ratio as a metric. The third is the **ordinal approach**, where the ordinal data is analysed in its original form. Within the ordinal approach, either a proportional odds model (frequentist or Bayesian)

**Table 1. Characteristics of included studies in the scoping review.**

| First author, year of publication | N[1] | | Ordinal scales | Type of reported data | Method (1) | Metric (1) | Method (2) | Metric (2) | The direction of effect size |
|---|---|---|---|---|---|---|---|---|---|
| Whitehead 1994 | 13 | | Different scales of endoscopic gastric damage (from 1 = "no ulcer" to 5 = "ulcer") (in one-point increments) | IPD[2] | Ordinal method (stratified proportional odds model **using maximum likelihood estimates** | Log-POR | Ordinal method (stratified proportional odds model **using binary split (Fisher's information)** | Log-POR | No change |
| White-head 2001 | 1 | 5 | CGIC[3] scale from 1 = " very much improved" to 7 = " very much worse" (in one-point increments) | IPD | Ordinal method (proportional odds model- **frequentist approach**) | Log-POR | Ordinal methods (proportional odds model-**Bayesian approach**) | BUGS mean | No change |
| | 2 | 8[4] | Different scales of endoscopic gastric damage (from 1 = "no ulcer" to 5 = "ulcer") (in one-point increments) | IPD | Ordinal method (proportional odds model- **frequentist approach**) | Log-POR | Ordinal method (proportional odds model-**Bayesian approach**) | BUGS mean | No change |
| Krishan 2013 | 216[5] | | mRS[6] from 0 = normal neurological state to 6 = death (in one-point increments) | IPD/SS[7] | Ordinal methods (**Proportional odds fixed effects model**) | Log-POR | Binary methods (**binary logistic regression fixed effects model**) | Log-OR | No change |
| Cumming 2015 | 5[8] | | EDSS[9] from 0 = normal to 10 = death (in half-point increments) | IPD/SS | Continuous method (**random-effects model**) | SMD | Ordinal method (**rank-based generalised model**) | Log-GeOR | No change |

[1]: number of studies included in the analysis.

[2]: individual patient data is used, i.e., the number of subject responses falling into each category for each treatment group is known.

[3]: Clinical General Impression of Change scale for Alzheimer's disease; however, in this set, the scale has been reduced into 5 categories combining grades 1 and 2, and grades 6 and 7.

[4]: these studies are a subset from Whitehead 1994.

[5]: from 24 meta-analyses.

[6]: modified Rankin Scale.

[7]: summary statistics outcome.

[8]: simulated studies.

[9]: Expanded disability status scale (EDSS) for multiples sclerosis. LogPOR: log proportional odds ratio. LogOR: log odds ratio. logGeOR: log generalised odds ratio. SMD: standardized mean difference.

or a generalised odds model is used to calculate the effect size, with the logarithmic proportional odds ratio (logPOR) or the logarithmic generalised odds ratio (logGeOR) as the respective metrics [15–17].

For the binary and continuous approaches, the included studies in the meta-analysis reported the ordinal outcomes either as summary statistics (such as means and standard deviations or counts/proportions) or as individual patient data (IPD), which provides the distribution of subject responses across ordered categories for each treatment group. Both reporting styles were suitable for these approaches.

In contrast, the ordinal approach requires individual patient data to perform the analysis. However, the generalised odds model can also be conducted using median and interquartile range (IQR) as summary statistics.

Table 1 summarises the methods with their metrics and the reported data type.

**Comparing the meta-analysis methods of ordinal data.** The comparison of meta-analysis methods for ordinal data revealed that the direction of the effect size remains consistent regardless of the approach used.

The earliest methodological study in this area was conducted by Whitehead and Jones [15], who utilised a proportional odds model to compare two techniques for calculating effect sizes: maximum likelihood estimates and Fisher's information. This study is summarised in Table 1.

A later study by Whitehead et al [16]. examined various models within the proportional odds framework, including comparing Bayesian and frequentist approaches.

The third study by Ashma Krishan and colleagues compared the binary approach with the proportional odds approach [14]. However, this study was published as a conference paper with limited details. Despite contacting the first and last authors, no full-text paper was available for further analysis.

The final study by Cumming, Churilov, and Sena reported a comparison between the continuous method and the ordinal generalised approach (generalised odds model) [17]. This comparison was based on mock data generated from an ordinal outcome scale, with only one iteration of a meta-analysis (as described in Table 1).

### Reproducible research analysis

One of the included studies [17], provided raw data, allowing us to conduct reproducible research. In this study, the authors used a neurological outcome scale, the Expanded Disability Status Scale (EDSS), to illustrate an ordinal approach (the rank-based generalised odds model) for meta-analysis of ordinal data and compared it with the continuous approach. The EDSS is used for multiple sclerosis and ranges from 0 (normal neurological state) to 10 (death), with half-point increments, except for the transition from 0 to 1, creating a total of 20 ordered categories [18].

To generate the data for their analysis, the authors used the EDSS distribution from a large observational study [19] to create a control group (sample 0) and five intervention groups (samples 1 to 5), each with 100 observations. They conducted a meta-analysis of five simulated studies, comparing each intervention group with the control group (e.g., sample 0 vs. sample 1, sample 0 vs. sample 2, etc.). The analysis was conducted using two approaches: calculating the standardised mean difference (SMD), representing the continuous approach, and calculating the log generalised odds ratio (logGeOR), representing the ordinal approach.

Their results showed statistically significant effect sizes in the same direction for both approaches (SMD = −0.17, 95% CI: −0.29 to −0.04; logGeOR = −0.21, 95% CI: −0.35 to −0.07). The heterogeneity was similar between the approaches ($I^2$ = 0% for both), as depicted in Figures 5 and 6 of the published report [17].

We attempted to reproduce these results using the raw data provided in the supplementary material, conducting meta-analyses with both SMD and logGeOR in two software platforms: STATA and R. Our results differed from the original study, showing statistically insignificant effect sizes in both STATA and R (SMD in STATA = −0.03, 95% CI: −0.16 to 0.09; logGeOR in STATA = −0.04, 95% CI: −0.19 to 0.10; SMD in R = −0.03, 95% CI: −0.15 to 0.10; logGeOR in R = −0.04, 95% CI: −0.19 to 0.10). The heterogeneity remained consistent across both platforms and approaches ($I^2$ = 0%). Fig 2 (for forest plots in STATA) and Fig 3 (for forest plots in R) illustrate these results.

To understand the discrepancies, we initially contacted the publishing journal (PLOS ONE), which referred us to the authors. We reached out to the authors via the corresponding email address provided (the last author), who then directed us to contact the first or second authors. Unfortunately, despite multiple attempts, we have not received a response.

To gain further insight into the discrepancies, we examined the raw data provided by the authors in the supplementary material and compared the summary statistics with those reported in the original study. This analysis revealed inconsistencies in three of the samples (Fig 4).

Despite these discrepancies, both analyses agreed on the direction of the effect size, and heterogeneity was similar between the two approaches. Further clarification from the original study's authors would be helpful, but our attempts to contact them were unsuccessful.

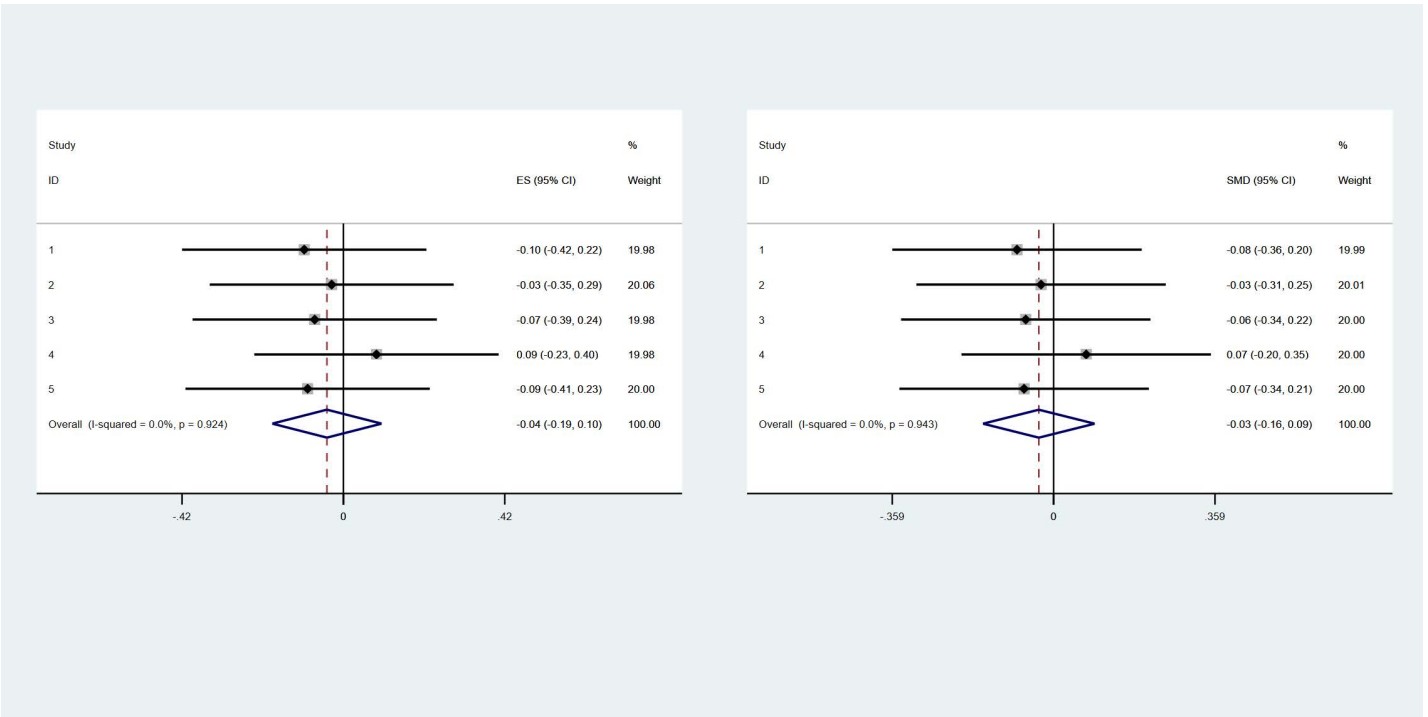

**Fig 2. Forest plots for our re-run of mocked-data in Cumming et al. 2015 using STATA.** Left using ordinal generalised odds model (with log GeOR as effect-size estimation) and right using continuous approach (with standardised mean difference as effect-size estimation).

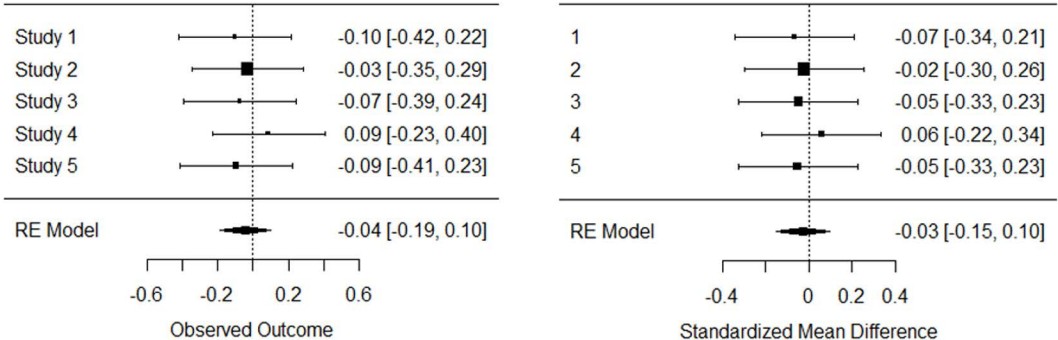

**Fig 3. Forest plots for our re-run of mocked-data in Cumming et al. 2015 using R.** Left using ordinal generalised odds model (with log GeOR as effect-size estimation) and right using continuous approach (with standardised mean difference as effect-size estimation).

## Discussion

### Summarising the literature

Despite more than thirty years since the foundation of the Cochrane Collaboration, meta-analysing ordinal data methods remain less well-established than binary and continuous data [1,2]. Our scoping review identified four methodological studies that addressed the meta-analysis of ordinal outcome scales with 5 to 20 ordered categories.

In 1994, Anne Whitehead and Nicola M. B. Jones proposed the stratified proportional odds model, combining multiple studies with differing classifications of ordinal outcomes. They

| Sample | N | Median (IQR) | Mann-Whitney U (vs sample 0) |
|--------|-----|-------------------|------------------------------|
| 0 | 100 | 6.0 (2.5, 6.5) | |
| 1 | 100 | 4.25 (2.0, 6.0) | 4155, z = -2.08 (p = .038) |
| 2 | 100 | 6.0 (2.5, 6.5) | 4782, z = -0.54 (p = .592) |
| 3 | 100 | 4.5 (1.625, 6.375) | 4186, z = -2.00 (p = .045) |
| 4 | 100 | 5.25 (2.125, 6.5) | 4490, z = -1.26 (p = .209) |
| 5 | 100 | 5.75 (2.5, 6.5) | 4764, z = -0.58 (p = .562) |

doi:10.1371/journal.pone.0145580.t001

| Sample | N | Median | 25th | 75th | Their | Mean |
|--------|-----|--------|------|------|---------|-------|
| 0 | 100 | 6 | 2,5 | 6,5 | Correct | 4,84 |
| 1 | 100 | 6 | 2,5 | 6,5 | False | 4,65 |
| 2 | 100 | 6 | 2,5 | 6,5 | Correct | 4,775 |
| 3 | 100 | 6 | 2,5 | 6,5 | False | 4,695 |
| 4 | 100 | 6 | 3 | 6,5 | False | 5,01 |
| 5 | 100 | 5,75 | 2,5 | 6,5 | Correct | 4,685 |

**Fig 4. Comparison of summary statistics of mocked data provided by Cumming et al. 2015.** The upper table is from their report, and the lower table is through our calculations (the discrepancies are in red).

used data from 13 trials examining the prevention of gastrointestinal damage with misoprostol, where the outcome scale ranged from 1 to 5, with one-point increments. This scale varied across studies, with some trials using only 2, 3, or 4 categories. They compared two ways of estimating logPOR and found no change in the direction of the effect size [15].

In a subsequent methodological study, Whitehead and colleagues introduced different proportional models to meta-analyse ordinal outcomes, including frequentist and Bayesian techniques [16]. They used individual patient data (IPD) for their analyses, allowing them to count the number of patients in each ordered category for each treatment group. The study included data from a subset of the same 13 trials from 1994 and another dataset from five trials evaluating the effect of tacrine in Alzheimer's disease patients. The outcome scale in the second dataset was the Clinical Global Impression of Change (CGIC), a seven-point scale with one-point increments [20]. This comparison also showed no change in the effect size between the two techniques of the proportional approach.

In 2013, Ashma Krishan and colleagues compared ordinal methods (using a proportional odds model) with binary methods (dichotomisation of the ordinal scale) using data from stroke trials in Cochrane systematic reviews [14]. The outcome scale was the modified Rankin Scale (mRS), which has seven ordered categories from 0 to 6 with one-point increments [21]. This study was available only as a conference paper, limiting its results' depth, which indicates no change in the effect-size direction.

Finally, Cumming, Churilov, and Sena used the Expanded Disability Status Scale (EDSS) for multiple sclerosis, where EDSS ranges from 0 to 10 with half-point increments, to present the rank-based generalised odds model as an ordinal approach and compared it with the continuous approach. They also found no change in the direction of effect size between the two approaches [17,18].

## Critical shortcomings in the existing literature

The existing methodological studies on the meta-analysis of ordinal outcomes have several shortcomings, limiting their capacity to provide robust empirical evidence for preferring one approach over another. The two studies by Anne Whitehead focused exclusively on different techniques and models within the proportional method, offering limited scope for broader comparisons. While the proportional odds method might be statistically appealing, in spite of its underlying assumptions [22], it requires effective implementation of raw data [16,22,23].

This reliance on individual patient data (IPD) makes it challenging to apply, as primary studies often do not provide sufficient data and there is currently no trend of complete sharing of raw data from clinical trials [24], which leads to the exclusion of studies from meta-analysis.

On the other hand, the binary and continuous approaches can be performed using either IPD or summary statistics, offering more flexibility [2]. However, the binary approach entails dichotomization, which in turn leads to loss of information [25], and the continuous approach entails the assumption of normality of distribution, which does not always hold [25,26]. The study by Ashma Krishan, which compared the proportional odds approach with the binary method, was published as a conference paper with incomplete reporting, raising concerns about the validity of its findings. And the study by Cumming and others attempted a comparison between the continuous approach and the generalised ordinal approach (generalised odds model). However, this analysis was based on only one simulated meta-analysis with no iterations. Moreover, our attempts to reproduce their results yielded different outcomes, casting doubt on the reliability of their conclusions.

## Strengths and limitations

In this review, we aimed to be comprehensive by conducting both electronic and citation searches, including screening the references of relevant studies. However, one limitation is that the screening and data extraction were performed solely by the main author due to the unavailability of other reviewers. Despite this, we believe this limitation did not significantly influence our findings, as the total number of citations screened was manageable for one reviewer.

A key strength of our review is our attempt to reproduce the results of previous studies, which is crucial for assessing the credibility of the evidence. However, we faced a limitation in this area as we were unable to contact the authors of the reproduced study to clarify the discrepancies in our results.

## Conclusions and implications

Given the current literature, a significant knowledge gap exists regarding the optimal method for meta-analysing ordinal outcome data, particularly for scales with 5 to 20 categories. Broadly, researchers have three main approaches to choose from: continuous, binary, and ordinal. However, the existing studies fail to provide robust empirical evidence to definitively support one approach over another, with many suffering from methodological limitations and a lack of comprehensive comparisons.

A new methodological study is crucial to addressing this gap. Such a study should aim to overcome the shortcomings of prior research and systematically compare the outcomes of meta-analyses for ordinal scales in clinical interventional trials across the three approaches: continuous, binary, and ordinal (including proportional and generalised models). To ensure reliability and reproducibility, the study should utilise clinical data and be validated through rigorous simulations with sufficient iterations.

## Supporting information

**S1 Checklist. PRISMA-ScR fillable checklist 10 Sept 2019.**
(DOCX)

## Author contributions

**Conceptualization:** Ali Mulhem.

**Data curation:** Ali Mulhem.

**Formal analysis:** Ali Mulhem.

**Methodology:** Ali Mulhem.

**Visualization:** Ali Mulhem.

**Writing – original draft:** Ali Mulhem.

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
