## [Decision Letter · Decision Letter 0]

16 Jan 2025

PONE-D-24-47567Meta-Analytical Approaches to Ordinal Outcome Data in Clinical Interventional Studies: A Scoping Review with Reproducible ResearchPLOS ONE

Dear Dr. Mulhem,

Thank you for submitting your manuscript to PLOS ONE. After careful consideration, we feel that it has merit but does not fully meet PLOS ONE’s publication criteria as it currently stands. Therefore, we invite you to submit a revised version of the manuscript that addresses the points raised during the review process.

We look forward to receiving your revised manuscript.

Kind regards,

Matthew Chin Heng Chua

Academic Editor

PLOS ONE

3. Please remove your figures from within your manuscript file, leaving only the individual TIFF/EPS image files, uploaded separately. These will be automatically included in the reviewers’ PDF.

Reviewers' comments:

Reviewer's Responses to Questions

**Comments to the Author**

1. Is the manuscript technically sound, and do the data support the conclusions?

Reviewer #1: Yes

2. Has the statistical analysis been performed appropriately and rigorously? 

Reviewer #1: Yes

3. Have the authors made all data underlying the findings in their manuscript fully available?

Reviewer #1: No

4. Is the manuscript presented in an intelligible fashion and written in standard English?

Reviewer #1: Yes

5. Review Comments to the Author

Reviewer #1: •The background information is insufficient, limiting a robust justification for why this research is needed and what it will add to the existing knowledge. 554938461.

•Methods: Can you explicitly state the framework used to guide the scoping review in the methods section?

•Methods: What specific approaches or thinking fed into the reproducible analysis?

•Results: The paragraph explaining the screening process is unclear; how did you arrive at the 108 papers, of which 104 were excluded?

•Results: Where are the included studies from, and which interventions did they focus on? Were the interventions similar across the included studies?

•Table 1, which is supposed to show the characteristics of the included studies, cannot be found in the manuscript or in any supporting file, making it difficult to appreciate the findings.

•Were the authors (those who provided the raw data) contacted regarding the discrepancies in the findings? If yes, what explanations were generated?

•What specific and critical research implications does your study offer for conducting meta-analysis with ordinal outcomes?

6. PLOS authors have the option to publish the peer review history of their article (what does this mean? ). If published, this will include your full peer review and any attached files.

**Do you want your identity to be public for this peer review?** For information about this choice, including consent withdrawal, please see our Privacy Policy .

Reviewer #1: No

---

## [Author Response · Author response to Decision Letter 1]

24 Jan 2025

A point-by-point response to reviewers

• Have the authors made all data underlying the findings in their manuscript fully available?

Reviewer #1: No

Response: Additional data have now been made available.

• Background information is insufficient, limiting a robust justification for why this research is needed and what it will add to the existing knowledge.

Response: A detailed section has been added to the introduction, providing a stronger justification for the research. It highlights the gaps in current knowledge, the importance of addressing these gaps, and the contributions this study makes to the field.

• Methods: Can you explicitly state the framework used to guide the scoping review in the methods section?

Response: The PRISMA framework has been explicitly mentioned under the “Scoping Review/Protocol and Registration” subsection to clarify the methodological guidance used.

• Methods: What specific approaches or thinking fed into the reproducible analysis?

Response: The primary aim and methodological approach of the reproducible analysis have been clarified in the “Reproducible Research” subsection of the Methods. We specified the rationale for verifying prior findings and addressing discrepancies.

• Results: The paragraph explaining the screening process is unclear; how did you arrive at the 108 papers, of which 104 were excluded?

Response: The paragraph describing the screening process has been rewritten for clarity. It now includes detailed steps on how the records were screened, excluded, and the reasons for exclusions.

• Results: Where are the included studies from, and which interventions did they focus on? Were the interventions similar across the included studies?

Response: The included studies are methodological studies focused on meta-analytical approaches for ordinal outcome data. As such, they do not address specific interventions but rather examine statistical methods. This is consistent with the stated eligibility criteria.

• Table 1, which is supposed to show the characteristics of the included studies, cannot be found in the manuscript or in any supporting file.

Response: Table 1 has now been included in the revised manuscript and supporting documents. We apologize for its omission in the initial submission.

• Were the authors (those who provided the raw data) contacted regarding the discrepancies in the findings? If yes, what explanations were generated?

Response: Yes, the authors were contacted. A detailed clarifying paragraph has been added under “Reproducible Research Analysis” in the Results section to address the discrepancies and the steps taken to resolve them.

• What specific and critical research implications does your study offer for conducting meta-analysis with ordinal outcomes?

Response: The critical implications have been elaborated on in the revised Conclusion and Implications section. This includes guidance on choosing statistical methods and the need for methodological rigor and reproducibility in future research.

---

## [Decision Letter · Decision Letter 1]

27 Feb 2025

Meta-Analytical Approaches to Ordinal Outcome Data in Clinical Interventional Studies: A Scoping Review with Reproducible Research

PONE-D-24-47567R1

Dear Dr. Ali Mulhem,

We’re pleased to inform you that your manuscript has been judged scientifically suitable for publication and will be formally accepted for publication once it meets all outstanding technical requirements.

Kind regards,

Yibeltal Alemu Bekele, MpH

Academic Editor

PLOS ONE

**Comments to the Author**

1. If the authors have adequately addressed your comments raised in a previous round of review and you feel that this manuscript is now acceptable for publication, you may indicate that here to bypass the “Comments to the Author” section, enter your conflict of interest statement in the “Confidential to Editor” section, and submit your "Accept" recommendation.

Reviewer #1: All comments have been addressed

2. Is the manuscript technically sound, and do the data support the conclusions?

Reviewer #1: Yes

3. Has the statistical analysis been performed appropriately and rigorously? 

Reviewer #1: Yes

4. Have the authors made all data underlying the findings in their manuscript fully available?

Reviewer #1: Yes

5. Is the manuscript presented in an intelligible fashion and written in standard English?

Reviewer #1: Yes

6. Review Comments to the Author

Reviewer #1: The authors have addressed all my comments satisfactorily. I recommended that the paper is accepted after a thorough grammar checks.

7. PLOS authors have the option to publish the peer review history of their article (what does this mean? ). If published, this will include your full peer review and any attached files.

**Do you want your identity to be public for this peer review?** For information about this choice, including consent withdrawal, please see our Privacy Policy .

Reviewer #1: No

---

## [Editor Report · Acceptance letter]

PONE-D-24-47567R1

PLOS ONE

Dear Dr. Mulhem,

I'm pleased to inform you that your manuscript has been deemed suitable for publication in PLOS ONE. Congratulations! Your manuscript is now being handed over to our production team.

Kind regards,

on behalf of

Mr. Yibeltal Alemu Bekele

Academic Editor

PLOS ONE